# Effects of Cassava Juice (*Manihot esculenta* Crantz) on Renal and Hepatic Function and Motor Impairments in Male Rats

**DOI:** 10.3390/toxins12110708

**Published:** 2020-11-09

**Authors:** Eduardo Rivadeneyra-Domínguez, José Eduardo Pérez-Pérez, Alma Vázquez-Luna, Rafael Díaz-Sobac, Juan Francisco Rodríguez-Landa

**Affiliations:** 1Facultad de Química Farmacéutica Biológica, Universidad Veracruzana, Xalapa 91000, Veracruz, Mexico; jeduardo_p1@hotmail.com (J.E.P-P.); almvazquez@uv.mx (A.V.-L.); radiaz@uv.mx (R.D.-S.); juarodriguez@uv.mx (J.F.R.-L.); 2Instituto de Ciencias Básicas, Universidad Veracruzana, Xalapa 91190, Veracruz, Mexico; 3Laboratorio de Neurofarmacología, Instituto de Neuroetología, Universidad Veracruzana, Xalapa 91190, Veracruz, Mexico

**Keywords:** cassava juice, renal function, hepatic function, motor incoordination, neurotoxic, hepatoxicity

## Abstract

Cassava (*Manihot esculenta* Crantz) is a plant that contains neurotoxins such as linamarin and lotaustraline. Its long-term consumption is associated with neuronal damage and contributes to the development of motor impairment in humans and rats. We investigated the effects of the consumption of cassava juice on renal and hepatic function and motor impairments in male rats. The rats received the vehicle, non-toxic and toxic doses of cassava juice, or linamarin as a pharmacological control, over 35 consecutive days. The effects were evaluated in an open field test, rotarod, and swim test. The toxic cassava dose and linamarin resulted in motor impairments in the rotarod and swim test from day 7 of treatment. The toxic cassava dose and linamarin increased the parameters that indicate renal and hepatic damage, with the exception of total protein and albumin levels. Behavioral variables that show motor incoordination (i.e., latency to fall in the rotarod) were negatively correlated with biochemical parameters of renal and kidney damage, whereas spin behavior was positively correlated. Our data indicate that chronic oral consumption of cassava juice caused renal and hepatic damage that was correlated with motor coordination impairment in rats, similarly to their principal neurotoxic compound, linamarin.

## 1. Introduction

Some plants that are used for food can contain toxic substances and can negatively impact health. The vast majority of vegetables contain substances that are nutritious and do not cause harm when consumed sporadically, but can be toxic in some cases, depending on the frequency of consumption, the amount ingested, and the vulnerability of individuals who consume them. Plants can also be toxic because of exogenous chemical contamination [1], which can negatively affect the function of diverse organs, including organs involved in detoxification (i.e., liver and kidney function) and the brain.

Cassava root (*Manihot esculenta* Crantz) is an alimentary source for a high percentage of people around the world, after rice, sugar, and corn. It has a wide geographic distribution and is easily cultivated. Cassava products have been introduced to the market in different regions of the world as part of the diet, despite containing more than one toxic substance [2]. The bromatological analysis of cassava tuber has reported the presence of water (62.2%), protein (1 %), fat (0.4%), total carbohydrates (32.8%), fiber (1%), ash (0.6%), calcium (40 mg), phosphorus (34 mg), iron (1.4 mg), thiamine (0.05 mg), riboflavin (0.04 mg), niacin (0.6 mg), ascorbic acid (19 mg), and an inedible portion (32%) in 100 g of fresh samples (Food and Agriculture Organization (FAO), 2008). Additionally, cassava tuber has a high concentration of cyanogenic glycosides, of which the most abundant is linamarin, followed by lotaustraline. Both these chemical compounds have been involved in the etiology of diverse motor alteration in humans and experimental animals.

The accumulation of cyanogenic glycosides depends on the genotype, the medium in which it is grown, agronomic practices, the age of the plant, and the part of the plant that is used [3]. In different geographic regions the excessive consumption of cassava has been consistently associated with some neuropathies. The consumption of improperly processed cassava derivatives, combined with an unbalanced diet that is deficient in sulfur amino acids, can lead to chronic cyanide poisoning [4]. Sulfur is essential in the cyanide detoxification process by converting cyanide to thiocyanate, which is eliminated in urine [5]. The consumption of fresh cassava or its derivatives has been associated with the development of neurological disorders, such as Konzo and tropical ataxic neuropathy [6]. Konzo is a neurological disorder that is characterized by selective damage to motor neurons, with an abrupt onset of irreversible, non-progressive, and symmetrical spastic paraparesis or tetraparesis [7]. Epidemiological studies have reported this disease in various rural areas in Africa [6], Mozambique [8], Tanzania [9], and the Democratic Republic of the Congo, particularly in growing areas where cassava is consumed as a food base [6]. Cases of Konzo also occur in other countries, but the true prevalence is unknown because it is often misdiagnosed as other central nervous system (CNS) disorders [10].

Preclinical research has found that the consumption of cassava in Wistar rats at a dose of 15 g/day increases the concentration of serum thiocyanate [11]. In another study in Wistar rats [12], the effects of aqueous and methanolic extracts of cassava were evaluated. Higher concentrations of alanine aminotransferase (ALT), aspartate aminotransferase (AST), alkaline phosphatase (ALP), urea, cholesterol, total proteins, and albumin were found. In addition, a decrease in white blood cells, lymphocytes, neutrophils, and hemoglobin was reported. Additionally, necrosis, the shrinking of glomeruli, and aggregates of lymphocytes were identified in the renal cortex, accompanied by the cytoplasmic vacuolization of hepatocytes and neurons. Cassava-fed rats gradually developed motor incoordination and exhibited lower dopamine levels in some brain areas, such as the striatum and cerebellum, both of which are involved in motor control [13,14]. Other studies reported the gradual development of motor incoordination, hyperactivity, a decrease in exploration, self-grooming behavior, and damage in such brain structures as the hippocampus [15].

These findings clearly indicate that the ingestion of cassava juice and exposure to active constituents of this plant consistently result in motor alterations and neuronal damage in brain structures that are involved in motor control and cognitive processes [16]. It has been suggested that these effects occur through the direct action of its components on the CNS, but it remains unknown whether other peripheral alterations at the level of metabolic processes also could contribute to neurological damage caused by cassava consumption. Although previous studies have identified biochemical changes associated with liver and kidney deterioration using methanolic extracts of cassava, it is unknown if total cassava tuber juice produces similar toxic effects to those of specific extracts of cassava on kidney and liver function, and if those changes are correlated with the motor coordination impairment associated with cassava consumption. The present study evaluated the effects of chronic cassava juice consumption on biochemical parameters that are indicators of renal and hepatic function, as well as the potential correlation with motor incoordination in rats. We sought to identify possible metabolic alterations that could contribute to motor impairment that is associated with cassava consumption.

## 2. Results

### 2.1. Body Weight

Table 1 shows the rats’ body weights throughout the study. The analysis of this variable did not show an effect of treatment (F_3,141_ = 1.829, *p* = 0.145) or interaction of factor (F_3,125_ = 0.202, *p* = 0.998). A significant effect of day was found (F_4,140_ = 32.821, *p* < 0.001). The post-hoc test showed that body weight significantly increased (*p* < 0.05) on days 14, 21, 28, and 35 compared with day 7, with significant differences between days 28 and 35 and days 14 and 21. A significant increase was observed on day 35 compared with day 28 (*p* = 0.005).

### 2.2. Behavioral Tests

#### 2.2.1. Open Field Test

##### Number of Crossings

No effect of treatment on the number of crossings was found (F_3,141_ = 2.543, *p* = 0.059), with no treatment × day interaction (F_3,125_ = 1.566, *p* = 0.110). A significant increase in the number of crossings was observed in the linamarin group compared with the other groups. A significant effect of day on the number of crossings was observed (F_4,140_ = 11.147, *p* < 0.001). A significant decrease in the number of crossings was observed on day 14 to day 35, compared with day 7 (see Table 2).

##### Time Spent Rearing

No effect of treatment on the time spent rearing was observed (F_3,141_ = 1.065, *p* = 0.366), with no treatment × day interaction (F_3,125_ = 0.820, *p* = 0.630). A significant effect of day on the time spent rearing was observed (F_4,140_ = 17.600, *p* < 0.001). A significant decrease in the time spent rearing was observed on days 14, 21, 28, and 35 compared with day 7 (*p* < 0.001; see Table 2).

##### Time Spent Grooming

Significant effects of treatment (F_3,141_ = 3.676, *p* = 0.014) and day (F_4,140_ = 5.439, *p* < 0.001) on the time spent grooming were observed, with no treatment × day interaction (F_3,125_ = 0.507, *p* = 0.907). The time spent grooming in the linamarin and cassava juice groups was significantly higher than in the control group (*p* < 0.05). A significant decrease in the time spent grooming was observed on days 28 and 35, compared with the earlier days of treatment (see Table 2).

#### 2.2.2. Rotarod

##### Latency to Fall

A significant effect of treatment on the latency to fall was observed (F_3,141_ = 27.014, *p* < 0.001). The linamarin group and toxic cassava dose group exhibited a shorter latency to fall (*p* < 0.05) compared with the control group and the non-toxic cassava dose group. A significant effect of day on the latency to fall was observed (F_4,140_ = 4.771, *p* = 0.001). A significant decrease in the latency to fall was observed on day 14 compared with day 7 (*p* < 0.05). No treatment × day interaction was observed (F_3,125_ = 0.488, *p* = 0.918; see Figure 1).

#### 2.2.3. Swim Test

##### Spin Behavior

Spin behavior was only displayed by rats in the linamarin group and the toxic cassava dose group (see Table 3). Thus, it was not possible to perform comparisons with the control group and non-toxic cassava dose group because spin behavior was zero in the latter two groups.

### 2.3. Biochemical Tests

#### 2.3.1. Renal Function

The biochemical indicators of renal function are shown in Table 4. Significant effects of group on glucose (F_3,25_ = 184.768, *p* < 0.001), creatinine (F_3,25_ = 1675.332, *p* < 0.001), urea (F_3,25_ = 43.777, *p* < 0.001), and blood urea nitrogen (BUN) (F_3,25_ = 129.665, *p* < 0.001) concentrations were observed. The values of these parameters significantly increased in the linamarin group and the toxic cassava dose group compared with the control group and the non-toxic cassava dose group (*p* < 0.05). The most pronounced effect on these parameters was found in the toxic cassava dose group.

#### 2.3.2. Liver Function

The biochemical indicators of liver function are shown in Table 5. Significant effects of group on total protein (F_3,25_ = 20.204, *p* < 0.001) and albumin (F_3,25_ = 17.353, *p* < 0.001) concentrations were observed. Significant decreases in these analytes were observed in the linamarin group and toxic cassava group compared with the control group and the non-toxic cassava group (*p* < 0.001). These effects were more pronounced in the toxic cassava group. Significant effects of group on total bilirubin (F_3,25_ = 65.62, *p* < 0.001), direct bilirubin (F_3,25_ = 28.953, *p* < 0.001), and indirect bilirubin (F3,25 = 98.987, *p* < 0.001) concentrations were observed. These analytes significantly increased in the linamarin group and the toxic cassava group compared with the control group and non-toxic cassava group. Significant effects of group on ALP (F_3,25_ = 23.854, *p* < 0.001), γ-glutamyl-transferase (γ-GT) (F_3,25_ = 1640.369, *p* < 0.001), AST (F_3,25_ = 130.668, *p* < 0.001), and ALT (F_3,25_ = 28.461, *p* < 0.001) concentrations were observed. A significant increase in these analytes was observed in the linamarin group and the toxic cassava group compared with the vehicle group and the non-toxic cassava group (*p* < 0.001).

### 2.4. Correlation between Biochemical Measures and Principal Behavioral Variables

We evaluated correlations between biochemical measures that denote liver and kidney damage and the principal behavioral variables that suggest motor coordination impairment (i.e., latency to fall in the rotarod and spin behavior in the swim test), in addition to crossing as a measure of horizontal spontaneous locomotion. No significant correlation was found between biochemical measures and crossing but, specifically, a significant negative correlation was found between latency to fall and creatinine, BUN, AST, ALT, ALP, total and indirect bilirubin, and γ-glutamyl-transferase. By contrast, a significant positive correlation was identified between spin behavior and glucose, urea, creatinine, BUN, AST, ALT, ALP, total and indirect bilirubin, and γ-glutamyl-transferase, whereas a significant negative correlation was found with direct bilirubin, total protein, and albumin (Table 6).

## 3. Discussion

Neurotoxicity studies have utilized experimental animal models to evaluate the effects of diverse substances on motor function and their relationship with CNS disorders [17,18,19]. Behavioral studies have evaluated specific patterns of motor behavior and biochemical, neurochemical, and molecular mechanisms that underlie these alterations, which can help explain the etiology of neurological disorders that are associated with the consumption of toxic substances [20].

In the present study, the treatments did not alter the variables in the locomotor activity test (i.e., crossing, rearing, and grooming) and only some effects were related with the time of treatment. These effects can be attributable to habituation processes that occurred under the repeated test conditions, in which the environmental conditions of the open field are recognized by the rats, thus resulting in a reduction of exploration, as previously reported [21,22,23]. Therefore, we could exclude possible effects of the treatments on horizontal spontaneous locomotor activity, exploration, and grooming under our experimental conditions.

The effects of the treatments on motor coordination were also explored in the rotarod test. This test is useful for assessing balance and motor coordination in rodents [24,25]. Healthy animals can balance on the rotarod for a longer time than animals with CNS alterations or animals under the influence of neurotoxins, which exhibit motor incoordination and quickly fall from the apparatus [26,27,28,29,30]. In the present study, the rats treated with linamarin and a toxic cassava dose quickly fell from the rotarod, thus indicating a motor coordination impairment, which is something that has been suggested to be associated with the neurotoxic compound content in cassava juice [31,32]. Additionally, results of the present study add new information that show a significant negative correlation between the latency to fall in the rotarod (i.e., an indicator of motor coordination impairment) and biochemical parameters that indicate a failure in liver and kidney function, which show that metabolic peripheral alterations could also contribute to the motor impairment associated with cassava juice consumption.

On the other hand, the swim test has been used to identify behavioral alterations that are associated with the consumption of toxic substances [15,33,34]. Spin behavior in this test is considered an indicator of motor incoordination in rats, in which the animal loses control of the motor coordination of its limbs, thereby preventing natural swim behavior [35]. This behavioral deterioration has been shown to occur in rats that were fed with cycad seeds (*Dioon spinulosum*) or microinjected in the dorsal hippocampus with neurotoxic metabolites of cycad (methylazoxymethanol) [33,34,35] or cassava (linamarin and acetonecyanohydrin) [31,32]. In the present study, the rats that were treated with linamarin and the toxic dose of cassava exhibited spin behavior, which was not detected in any of the rats treated with vehicle or non-toxic doses of cassava. This finding confirms the motor coordination impairment in the rats associated with the cassava juice consumption. Previous studies have suggested that impaired motor coordination is likely associated with the toxic action of cyanide that is derived from linamarin content in cassava juice during their biodegradation [36,37,38]. Notably, cyanide that is derived from cassava causes the demyelination of spinal cord neurons, which is related to a lack of limb coordination [8]. The daily consumption of cyanide products, such as those from cassava, has been linked to such neurological disorders as tropical ataxic neuropathy and Konzo [15]. In addition, the results of the present study add new information that show a significant positive correlation between spin behavior (i.e., an indicator of motor incoordination) and biochemical parameters that indicate liver and kidney damage, suggesting that metabolic alterations may contribute to the motor impairment associated with cassava juice consumption in the long-term. Indeed, hepatotoxicity implies liver lesions, which are caused by exposure to diverse substances [39]. Any infectious, degenerative, neoplastic, or toxic process that exceeds the functional capacity of the liver can cause hepatic failure [40]. To detect abnormal liver function, biochemical parameters are evaluated in serum or plasma samples, including bilirubin, total proteins, albumin, AST, ALT, ALP, and γ-GT [41]. Increases in serum concentrations of ALT, AST, and ALP, as well as higher levels of total and conjugated bilirubin, are biological markers of hepatic damage. The levels of total bilirubin and direct bilirubin reflect the liver’s ability to pass bilirubin from plasma to bile. Clinical studies have demonstrated that liver failure associated with the consumption of alcohol can lead to hepatic encephalopathy, which is associated with motor incoordination and deterioration of cognitive and memory processes [18]. In this way, the modification of biochemical parameters related to kidney and liver damage associated with cassava juice consumption suggest that motor incoordination in cassava consumers may be influenced by peripheral metabolic changes.

The biochemical analysis in rats that were treated with linamarin and cassava juice revealed higher concentrations of AST, ALT, γ-GT, ALP, and bilirubin and lower concentrations of total proteins and albumin, which typically result from hepatic damage [12,32,42,43]. In this way, it is probable that the constituents of cassava juice and linamarin cause vacuolar degeneration in the liver because of a high content of cyanogenic glycosides, which consequently increases the concentration of cyanide. This increase affects the cytochrome P450 enzymatic systems, causing the blockade of cellular oxygenation and producing anoxia and subsequent hepatic vacuolar degeneration [44]. In addition, elevated transaminase concentrations are associated with diverse hepatopathies. Levels of AST are associated with disorders of the heart and skeletal muscles, whereas alanine aminotransferase is considered a more specific marker of chronic or infiltrative liver disease. Higher concentrations of ALP and γ-GT are indicators of liver disease [45], whereas a decrease in total proteins and albumin is indicative of chronic liver disease or abnormal excretion in some nephropathies [46]. In our study, similar results were observed in rats that were treated with cassava juice or linamarin, suggesting the potential establishment of hepatotoxicity, which could contribute in the motor incoordination found here. In support of this, negative or positive correlations were found between biochemical measures suggestive of kidney and liver damage and behavioral indicators of motor incoordination.

The kidneys eliminate, among other products, urea, uric acid, and creatinine, in addition to metabolizing and eliminating drugs and toxins [41]. Acute renal injury is characterized by increases in nitrogen waste products, such as urea nitrogen and creatinine, in addition to glucose, urea, creatinine, and BUN, which are retained in the blood [47] and which are indicators of glomerular filtration failure [42,48,49]. Therefore, a decrease in urinary volume causes an increase in the passive reabsorption of urea and a decrease in its elimination, which depends on protein intake and catabolism [50]. This failure in the detoxification processes could have occurred in the present study, in which urea and creatinine, among the other biochemical parameters, were higher in the rats treated with linamarin and the toxic cassava juice dose, suggesting a decrease in glomerular filtration [51]. It is possible that failure in the elimination of potential toxic substances related to kidney and liver damage associated with cassava juice or linamarin consumption could contribute to some neurological alteration and then deteriorate motor coordination in the rats. As mentioned, this is supported by the correlational study, in which a significant negative correlation was found between latency to fall (i.e., deterioration in motor coordination) and creatinine, BUN, AST, ALT, ALP, total and indirect bilirubin, and γ-glutamyl-transferase. A significant positive correlation between spin behavior (i.e., deterioration in motor coordination) and glucose, urea, creatinine, BUN, AST, ALT, ALP, total and indirect bilirubin, and γ-glutamyl-transferase was also found, along with a negative correlation with direct bilirubin, total protein, and albumin. Interestingly, none of the biochemical variables was correlated with crossing (i.e., horizontal spontaneous locomotion), which highlights specific damage in motor coordination, but not in general locomotion. In support of this, it has been reported that the increases in the biochemical parameters here evaluated are directly related with motor incoordination and memory deterioration. For example, hepatic neuropathy associated with the consumption of alcohol increases the concentration of AST, ALT, ALP, total and indirect bilirubin, and γ-glutamyl-transferase due to liver injury. These substances cross the blood–brain barrier, producing neurotoxicity associated with deterioration of cognitive and memory process, in addition to motor incoordination, which is a characteristic of hepatic neuropathy [39].

Finally, the present study has limitations. First, the cassava juice used in the present research was not chemically characterized to quantify the content of linamarin; however, although we cannot discount different concentrations in the chemical compounds of the cassava juice, it is evident that the behavioral effects associated with cassava juice are produced consistently with the same doses and the same processes of extraction of the cassava juice [15,32,52]. In fact, in one study using cassava juices prepared under the same conditions but extracted from cassava from another geographic region, the same motor effects were found [53]. This suggests that independently of the origin of cassava tuber, it maintains its toxic effects on motor coordination and biochemical parameters in rats. Second, in the present research the biochemicals measured in the blood were considered to be indirect biomarkers of kidney and liver damage, but histological analysis was not performed. Nonetheless, other studies have histologically evaluated the liver damage produced by cassava or its constituents [12]. In this way, the contribution of the present study shows the correlation between biochemical parameters suggestive of liver and kidney failure with the motor incoordination in rats treated with cassava juice, which probably is related to their content of linamarin, considering that in the present study similar correlations and effects were found in rats treated with linamarin.

## 4. Conclusions

The chronic oral administration of cassava juice caused kidney and liver damage that was correlated with motor impairment in rats, which could contribute to the understanding of etiology of motor incoordination reported in some cassava consumers. Future studies need to evaluate the histopathology of the kidneys, liver, and brain to identify specific structural damage that is caused by cassava consumption.

## 5. Materials and Methods

### 5.1. Animals

The study included 29 adult male Wistar rats, weighing 250–300 g at the beginning of the experiments. The rats were housed in Plexiglas cages (5 rats per cage) under a 12 h/12 h light/dark cycle (lights on at 7:00 am) and average room temperature of 25 °C ± 2 °C. The animals had ad libitum access to water and food.

### 5.2. Ethical Approval

The experimental protocols were strictly performed according to the Guide for the Care and Use of Laboratory Animals [54] and Official Mexican Standard NOM-062-ZOO-1999 [55]; additionally, the recommendation stated by the 3 Rs of Russell (Reduce, Replace and Refine) as applied to experimental research in animals were considered [56]. The welfare of the rats was checked daily according to the rat grimace scale [57]. Body weights were recorded every three days. The protocol was approved by the Internal Committee for the Care and Use of Laboratory Animals of the Institute of Health Sciences (CICUAL-ICS) with registration number 2018-002B, dated 18 February 2019.

### 5.3. Biological Material

All cassava tubers (*M. esculenta* Crantz) used in the present study were collected from the same site and the same season to minimize potential variation in the phytochemical profile, as has been done in previous studies [15,32,52]. Under this condition, cassava juice produces the motor impairment associated with neuronal damage in the rat hippocampus. Cassava tubers were free of agrochemicals, and were collected using traditional cultivation methods in the town of Defensa, Yecuatla county, Veracruz state, México (latitude: 19°52′00″ N; longitude: 96°47′00″ W) at an altitude of 260 m above sea level. The authentication of the biological material was performed in the Herbarium XAL at the Institute of Ecology A.C. (INECOL) in Xalapa city by taxonomist Sergio Avendaño Reyes.

### 5.4. Cassava Juice

The extraction of cassava juice was performed according to previous studies [52]. Every day, before administration, fresh cassava tuber juice was obtained with a juice extractor machine (Moulinex Model Centri III, Celaya, Guanajuato, México) and immediately administered to the rats according to correspondent doses.

In the present study, the phytochemical profile of cassava juice was not determined. In previous studies, high-performance liquid chromatography (HPLC) was used to estimate the content of linamarin in different doses that are effective or ineffective to produce motor impairment [15]. Subsequently, using similar doses of cassava juice extracted under the same conditions described there, it was possible to identify motor impairment and neuronal damage in the hippocampus and other brain structures [32,52,53], which suggests that the linamarin content in similar doses of cassava juice under the same processes of extraction could be similar.

### 5.5. Dose Selection

In previous studies it was determined by HPLC that cassava juice obtained from 3.5 g of cassava tuber contains approximately 0.074 mg linamarin and was ineffective to produce motor impairment when it was injected per kilogram of rat weight; whereas cassava juice obtained from 28.5 g of cassava tuber contained approximately 0.3 mg linamarin, which was effective to produce motor impairment when was injected per kilogram of rat weight [15,52]. To standardize the volume of cassava juice administered and avoid any experimental artifacts, all of the treatments were adjusted to a final volume of 2 mL per kilogram of the rats’ weights, adding purified water. Under these considerations, in the present study, the non-toxic dose and toxic dose of cassava juice (containing approximately 0.75 and 0.3 mg of linamarin, respectively) were used for further testing. Then, linamarin at 0.3 mg/kg (considering that this is the quantity of linamarin contained in the juice obtained from 28.5 g of cassava tuber, the toxic dose) was selected as the pharmacological control, considering that this dose produces neurotoxicity and motor impairment in rats [15].

### 5.6. Experimental Groups

The present study included four independent groups of rats (*n* = 6–8/group)—a control group that received purified water (considering that purified water was used to standardize cassava juice to 2 mL and it was the vehicle used to dissolve linamarin), two groups that were treated with the cassava juice obtained from 3.57 (non-toxic doses) and 28.56 g (toxic doses) of cassava tuber and dosed per kilogram of rat weight, and one group that was treated with linamarin (0.3 mg/kg; Sigma-Aldrich, St. Louis, MO, USA). The treatments were administered orally every 24 h over 35 consecutive days. The effects of the treatments were evaluated every 7 days until day 35 in a battery of behavioral tests, starting with the open field test, followed by the rotarod, and finally the forced swim test. For the behavioral tests; approximately, 5 min elapsed between each test. On day 35 of the treatment, 5 min after the forced swim test, the rats were anesthetized with sodium pentobarbital (90 mg/kg, i.p.; Cheminova de México, Mexico City, Mexico; Reg. SAGARPA Q-7048-044) to obtain a blood sample by cardiac puncture to quantify the biochemical parameters.

### 5.7. Behavioral Tests

The rats were evaluated in a behavioral test battery that began with the open field test, followed by the rotarod test, and then the swim test. Approximately 5 min elapsed between tests.

#### 5.7.1. Open Field Test

The rats were individually placed in an opaque Plexiglas cage (44 cm × 33 cm base, 20 cm high). The floor was delineated into 12 squares (11 cm × 11 cm). The following variables were evaluated: (a) number of crossings, when the rat passed at least three-quarters of its body from one square to another; (b) time spent rearing, when the rat assumed a vertical posture relative to the floor, supported on its hind limbs; and (c) time spent self-grooming, an indicator of motivational state of the animal [33]. After this test, the rats were evaluated on the rotarod.

#### 5.7.2. Rotarod

The rats were trained on a rotarod (LE 8300, Letica LSI, Panlab Scientific Instruments, Barcelona, Spain) for 5 consecutive days at a speed of 18 rotations per minute before the treatments were given. The latency to fall from the rotarod was recorded, which was the time that elapsed after the rat was placed on the rod until it fell. This test was conducted 0, 7, 14, 21, 28, and 35 days after treatment to identify changes in motor coordination and balance [58]. After the rotarod session, the rats were evaluated in the swim test.

#### 5.7.3. Swim Test

The rats were individually placed in a rectangular pool (26 cm × 29 cm × 50 cm) that was filled with water (25 ± 1 °C) for 5 min. The depth of the water was adjusted so that the rat could touch the bottom of the pool with one or both of its hindlimbs and tail. None of the animals drowned. The variable that was evaluated in this test was spin behavior, which is considered an indicator of motor incoordination [15]. Spin behavior was defined as periods in which the rat swam on its own axis, without horizontal displacements [15,33].

All of the locomotor activity and swim test sessions were videotaped. Two blind independent observers quantified the variables until they reached a consensus of at least 95%. Spin behavior in the swim test was evaluated in the videos and automatically analyzed using ANYmaze 4.73 software (Stoelting, Wood Dale, IL, USA).

### 5.8. Blood Samples

After the last behavioral test, blood samples were obtained. We used 5 mL syringes with a 22 mm needle length. The rats were anesthetized with sodium pentobarbital (90 mg/kg, i.p., Cheminova de México, Mexico City, Mexico; Reg. SAGARPA Q-7048-044). They were then placed in the supine position and the syringe was inserted through the lateral thoracic wall and intercostal space in the maximum region of the heartbeat at an angle of 20° to 30°. The needle was then moved slowly, thus making slight negative pressure in the cylinder of the syringe, and then carefully drawn until blood flow stopped [59]. The blood sample was deposited in Vacutainer tubes (BD Vacutainer, Mexico City, Mexico) without anticoagulant (dry). The blood samples were allowed to coagulate and then centrifuged at 3500 rotations per minute for 5 min to obtain the serum, which was immediately transferred with a Pasteur pipette to corresponding containers for dry chemistry analysis using a Vitros250 device (Johnson and Johnson, Ramsey, MN, USA). Glucose, creatinine, urea, blood urea nitrogen (BUN), total proteins, albumin, total bilirubin, direct bilirubin, indirect bilirubin, ALP, γ-GT, AST, and ALT were analyzed. Finally, the normal reference intervals of the tests (see Table 7) were compared to verify possible renal and hepatic alterations According to Suckow [60] and Sharp and Villano [61].

### 5.9. Statistical Analysis

The behavioral and body weight data were analyzed using two-way repeated-measures analysis of variance (ANOVA). The biochemical parameters were analyzed using one-way ANOVA. When significant differences in the ANOVA were attained, the Student–Newman–Keuls post-hoc test was performed. The data are expressed as mean ± standard error of the mean (SEM). Finally, correlations between the principal behavioral variables associated with motor incoordination (i.e., crossing, latency to fall, and spin behavior) and biochemical parameters indicative of liver and kidney failure were investigated using Pearson’s correlation analysis. The statistical analysis was performed using SigmaStat 3.5 software (SAS Institute Inc., Cary, NC, USA).

## Figures and Tables

**Figure 1 toxins-12-00708-f001:**
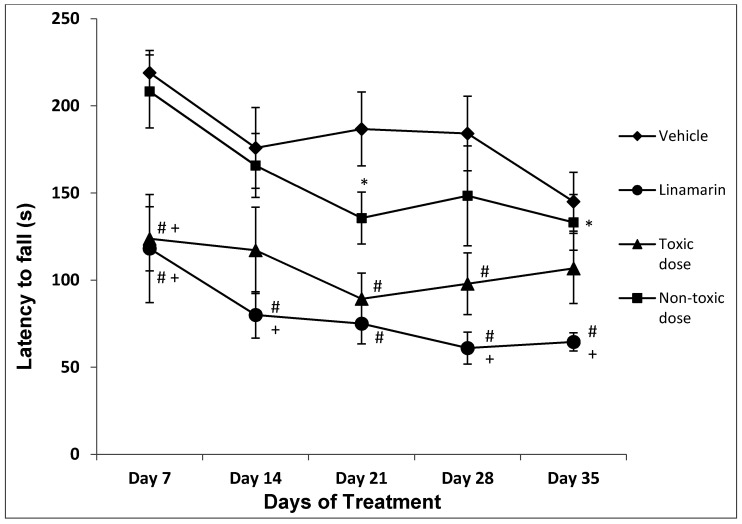
Latency to fall. A significant decrease of this behavior was shown in the linamarin group with respect to the control and non-toxic dose groups; the toxic dose group significantly decreased this behavior on days 7, 21 and 28 with respect to the control and on day 7 with respect to the non-toxic dose group. * *p* < 0.05 vs. Day 7; # *p* < 0.05 vs. Control; + *p* < 0.05 vs. Non-toxic dose. ANOVA two-way, post hoc Student–Newman–Keuls.

**Table 1 toxins-12-00708-t001:** Body weight of the rats according to treatment, days of treatment and interaction of factors.

Treatments					
Treatment Days	Control	Linamarin	Toxic Doses	Non-Toxic Doses	Treatment Days Factor
7	316 ± 4	316 ± 9	314 ± 6	322 ± 6	317 ± 3
14	335 ± 4	327 ± 11	333 ± 6	339 ± 7	334 ± 3 *
21	344 ± 5	329 ± 18	348 ± 5	351 ± 6	344 ± 4 *
28	357 ± 5	353 ± 10	361 ± 6	363 ± 7	359 ± 3 *^,#^
35	372 ± 5	371 ± 11	377 ± 5	377 ± 7	374 ± 3 *^,#,$^
Treatment factor	345 ± 4	339 ± 6	346 ± 4	350 ± 4	

Values are expressed as the mean ± standard error. * *p* < 0.05 vs. day 7; ^#^
*p* < 0.05 vs. 14 and 21 days; ^$^
*p* = 0.005 vs. day 28. Analysis of variance (ANOVA) two-way, post hoc Student–Newman–Keuls.

**Table 2 toxins-12-00708-t002:** Variables evaluated in the open field test.

Treatments						
Test	Treatment Days	Control	Linamarin	Toxic Doses	Non-Toxic Doses	Total
Crossing (n)	7	48.6 ± 4.7	57.8 ± 4.0	54.4 ± 6.4	60.1 ± 5.2	55.3 ± 2.6
	14	41.6 ± 5.1	38.6 ± 6.4	52.9 ± 3.4	45.1 ± 7.4	45.1 ± 2.9 *
	21	33.5 ± 6.8	32.4 ± 3.9	29.4 ± 5.1	31.2 ± 5.9	31.5 ± 2.7 *
	28	25.4 ± 2.3	60.2 ± 10.2	30.7 ± 5.9	36.6 ± 7.1	37.1 ± 3.9 *
	35	27.0 ± 5.0	40.7 ± 9.3	27.6 ± 2.9	31.6 ± 5.9	31.3 ± 2.9 *
	Total	35.2 ± 2.6	45.9 ± 3.6	39.0 ± 2.8	40.9 ± 3.2	
Rearing (s)	7	65.3 ± 8.6	63.7 ± 9.7	65.8 ± 6.8	63.4 ± 14.8	64.6 ± 5.0
	14	40.4 ± 11.1	39.2± 7.2	59.6 ± 5.8	38.1 ± 7.0	44.8 ± 4.1 *
	21	21.8 ± 7.0	41.5 ± 7.4	38.5 ± 4.0	33.1 ± 4.6	33.6 ± 3.0 *
	28	20.8 ± 4.4	22.9 ± 8.8 *	31.6 ± 6.1	30.6 ± 4.3	26.9 ± 2.9 *
	35	35.8 ± 4.7	28.4± 6.9 *	26.4 ± 6.1	26.6 ± 4.2	29.1 ± 2.3 *
	Total	36.8 ± 4.2	39.2 ± 4.2	44.4 ± 3.4	38.4 ± 3.9	
Self-grooming (s)	7	32.8 ± 7.1	72.1 ± 3.8	56.1 ± 13.8	71.4 ± 13.4	58.0 ± 6.1
	14	44.0 ± 19.5	68.4 ± 19.1	45.6 ± 12.8	43.7 ± 12.9	49.4 ± 7.7
	21	22.7 ± 12.8	51.2 ± 15.4	61.7 ± 16.0	60.2 ± 12.0	49.7 ± 7.3
	28	13.1 ± 4.7	28.2 ± 7.6	30.8 ± 15.0	36.2 ± 10.7	27.5 ± 5.4 *
	35	13.5 ± 4.8	26.7 ± 12.0	30.5 ± 14.2	27.5 ± 9.4	24.7 ± 5.3 *
	Total	25.2 ± 5.1	49.3 ± 6.4 ^+^	44.9 ± 6.4 ^+^	47.8 ± 5.6 ^+^	

Values are expressed as the mean ± standard error. n, number; s, seconds. * *p* < 0.05 vs. day 7. ^+^
*p* < 0.05 vs. control group. ANOVA two-way, post hoc Student–Newman–Keuls.

**Table 3 toxins-12-00708-t003:** Variable evaluated in the swim test *.

Treatments					
Test	Treatment Days	Control	Non-Toxic Doses	Linamarin	Toxic Doses
Spin behavior (n)	7	0	0	2.8	3.4
	14	0	0	3.0	6.2
	21	0	0	5.0	6.5
	28	0	0	5.6	6.3
	35	0	0	9.2	8.5

* Not statistical analysis was performed in this variable considering that it was only shown in two groups of treatment, but not in the control group, which evidently is significant.

**Table 4 toxins-12-00708-t004:** Effects of oral administration of cassava juice and linamarin on renal function in the rat.

Treatment				
Analyte	Control	Linamarin	Toxic Dose	Non-Toxic Dose
Glucose (mmol/L)	6.0 ± 0.2	7.6 ± 0.1 *	9.4 ± 0.1 *	6.2 ± 0.1
Urea (mmol/L)	16.8 ± 1.1	21.7 ± 0.6 *	29.8 ± 1.2 *	18.0 ± 0.2
Creatinine (µmol/L)	10.1 ± 0.5	58.8 ± 0.8 *	66.3 ± 1.0 *	13.2 ± 0.2 *
Blood Urea Nitrogen (mmol/L)	6.3 ± 0.1	8.1 ± 0.2 *	8.6 ± 0.1 *	6.2 ± 0.1

Values are expressed as the mean ± standard error. * *p* < 0.05 versus the control group. One-way ANOVA for independent groups followed by the Student–Newman–Keuls post hoc test.

**Table 5 toxins-12-00708-t005:** Effects of oral administration of cassava juice on liver function in the rats.

Treatment				
Analyte	Control	Linamarin	Toxic Dose	Non-Toxic Dose
Aspartate Amino Transferase (UI/L)	93.80 ± 1.49	158.25 ± 10.51 *	188.26 ± 1.93 *	79.81 ± 2.26 *
Alanine Amino Transferase (UI/L)	49.21 ± 2.39	67.08 ± 1.42 *	67.47 ± 1.04 *	43.17 ± 3.38
Alkaline Phosphatase (UI/L)	242.85 ± 10.27	381.66 ± 2.69 *	422.62 ± 33.15 *	234.00 ± 10.69
Total Bilirubin (mg/dL)	0.39 ± 0.01	0.67 ± 0.01 *	0.67 ± 0.01 *	0.39 ± 0.01
Indirect Bilirubin (mg/dL)	0.02 ± 0.01	0.23 ± 0.01 *	0.19 ± 0.02 *	0.01 ± 0.01
Direct Bilirubin (mg/dL)	0.01 ± 0.01	0.14 ± 0.01 *	0.25 ± 0.03 *	0.05 ± 0.02
Total Proteins (g/dL)	5.38 ± 0.04	3.66 ± 0.39 *	4.33 ± 0.10 *	5.32 ± 0.08
Albumin (g/dL)	4.15 ± 0.06	2.96 ± 0.29 *	3.10 ± 0.16 *	4.16 ± 0.05
γ-Glutamyl-Transferase	23.28 ± 0.86	83.50 ± 1.25 *	95.25 ± 1.19 *	23.62 ± 0.37

Values are expressed as the mean ± standard error. * *p* < 0.05 versus the control group, One-way ANOVA for independent groups followed by the Student–Newman–Keuls post hoc test.

**Table 6 toxins-12-00708-t006:** Correlation coefficients of biochemical measures and principal behavioral variables.

Biochemical Measures	Behavioral Variables		
	Crossing (n)	Latency to Fall (s)	Spin Behavior (n)
**Renal function**			
Glucose (mmol/L)	r −0.001	r −0.304	r 0.673 *
Urea (mmol/L)	r −0.033	r −0.299	r 0.594 *
Creatinine (µmol/L)	r 0.072	r −0.012 *	r 0.793 *
Blood Urea Nitrogen (mmol/L)	r 0.020	r −0.036 *	r 0.687 *
**Liver function**			
Aspartate Amino Transferase (UI/L)	r 0.024	r −0.384 *	r 0.078 *
Alanine Amino Transferase (UI/L)	r 0.038	r −0.372 *	r 0.633 *
Alkaline Phosphatase (UI/L)	r 0.092	r −0.415 *	r 0.644 *
Total Bilirubin (mg/dL)	r 0.118	r −0.398 *	r 0.738 *
Indirect Bilirubin (mg/dL)	r 0.170	r −0.434 *	r 0.792 *
Direct Bilirubin (mg/dL)	r −0.211	r 0.354	r −0.568 *
Total proteins(g/dL)	r −0.211	r 0.354	r −0.568 *
Albumin (g/dL)	r −0.268	r 0.312	r −0.633 *
γ-Glutamyl-Transferase	r 0.083	r −0.445 *	r 0.804 *

(n), number; (s), time in seconds; r, Pearson’s correlation coefficient. Significance taken at *p* < 0.05, as denoted by *.

**Table 7 toxins-12-00708-t007:** Reference intervals for evaluated analytes indicative of renal and hepatic function in Wistar rats.

Analyte	Reference Intervals
Glucose	6–10 mmol/L
Creatinine	11–28 µmol/L
Urea	10.7–20 mmol/L
Blood Urea Nitrogen	3–7 mmol/L
Total proteins	5–7 g/dL
Albumin	4–5 g/dL
Total bilirubin	0.18–0.54 mg/dL
Direct bilirubin	0.03–0.06 mg/dL
Indirect bilirubin	0–0.1 mg/dL
Alkaline Phosphatase	36–312 UI/L
γ-Glutamil-Transferase	8.8–24 UI/L
Aspartate Amino Transferase	63–157 UI/L

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
