# Peer review of "Effects of Cassava Juice (Manihot esculenta Crantz) on Renal and Hepatic Function and Motor Impairments in Male Rats"

_toxins, 2020, doi:10.3390/toxins12110708_

Round 1
Reviewer 1 Report
In this manuscript, the authors describe a series of studies to evaluate the toxicity of a cassava extract in rats, including hepatotoxicity, nephrotoxicity, and behavioral and motor coordination effects.
Overall, I felt that there was very little novelty to this study. The authors indicate over and over in the manuscript that various aspects of the toxicity of cassava have already been reported in the literature. At the end of reading the manuscript, I was left wondering what new data was presented.
I felt that the evaluation of the hepatotoxicity and nephrotoxicity was very cursory with no obvious data indicating how liver and kidney injury would affect neurotoxicity of the plant. This needs to be explained in much greater detail. The addition of histology would greatly improve the quality of this manuscript. The changes in serum ALT and AST were very minimal. Although they may be statistically significant, I would argue that they are not biologically significant. In the field of drug induced liver injury, an increase in serum ALT is not considered important until it is 3 times that of normal.
The authors repeatedly refer to metabolic products of cassava, but they show no data for metabolic products nor any description of methods as to how this was done. I wonder if the authors have intended another meaning for metabolic products.
The first sentence of the conclusion is a gross overstatement of the results presented in the manuscript and thus needs to be reworded.
Minor comments:
The authors need to review the rules for reporting significant numbers. the presentation of the data in the tables could be cleaned up considerably by reporting fewer significant numbers. For example, for Table 1, the data should reported as whole numbers.
Please select new symbols for the figure legend for figure #1. It is hard to tell which group is which. As it is right now, it appears that linamarin group had little effect.
Tables 4&5 there is no designation in the table for which groups are significantly different. Table 5, spell albumin and bilirubin correctly.
If the cassava juice extract was prepared fresh each day, how do you know that the animals received the same dose each day? There could have been daily variations in the efficiency of the extraction. Also, it is not clear what the units g/kg refer to. grams of what?
why was the control group provided purified water?
Were the rats allowed a period of rest between assays, or were they moved from to the next assay immediately after one finished?
Reviewer 2 Report
Please see my comments in the attached file.

Author Response
Please, see the attachment.

Reviewer 3 Report
This is a paper that analyzes the effect of chronic consumption of cassava on Wistar rats, which contains neurotoxins such as linamarin,
After evaluating its effect on biochemical parameters and liver and kidney function, alterations were observed mainly with linamarin and with the toxic fraction of cassava. What had a direct impact on the central nervous system of animals, this after having analyzed the behavior through various tests
I consider that the work protocol is adequate as well as the results presented which support the conclusions.
Author Response
Please, see the attachment.

Reviewer 4 Report
The work describes in a fairly detailed way the experiments on animals - which are of course the essence of this work and it was on their basis that the presented conclusions were drawn, but almost no attention was paid to the biochemical analysis of the plant material - I mean the analysis of cassava juice used in the experiments.
The work cannot be published without supplementing this information. Please answer the following questions:
1. Was the content of linamarine in the juice used for the experiments assessed and how (toxic and non-toxic dose)?
2. How does this content correlate with the linamarine experiment - which test does this experiment correspond to?
3. What other substances are in this juice - sugars, phenolic compounds, terpenoid compounds, vitamins - antioxidant compounds, potential detoxifiers, enzymes involved in the metabolism of cyanogenic glycosides, etc. The presence of such substances may affect the test result - did the authors conduct such an analysis - have such data from other scholars are available - at least discuss it.
4.I understand that the juice was always obtained from the same mass of cassava (but different plants / biological samples) - whether the toxic substances are distributed in the same tissue, similar in different plants - or this guarantees the consistency of doses - has it been checked in any way - biological material can be diversified - one plant (tissue) of another is not equal - which can generate "false" experimental differences.
Author Response
Please, see the attachment.

Round 2
Reviewer 1 Report
better
Reviewer 2 Report
The authors resolved most of my concerns.
However, Table 6 should be re-performed as follows:
- Table caption should be: The correlation coefficient of biochemical measures and principal behavioral variables
- Table content: No need to put "r" and "p". Just keep r-value, remove p-value. Keep the "*" symbol along with r-value. Because you already said "significance taken at p < 0.05, 193 as denoted by *"
- mg/dI or mg/dL?
Also, please carefully check the English spelling.
Reviewer 4 Report
I am not entirely satisfied with the information on the biochemical composition of the cassava juice used, but after completing the article with information from previous research and pointing to potential imperfections in the discussion and description of the results, I am able to accept the article in its current form :).
